# A two–tiered system for selective receptor and transporter protein degradation

**Charlotte Kathleen Golden, Thomas David Daniel Kazmirchuk** **, Erin Kate McNally, Mariyam El eissawi, Zeynep Derin Gokbayrak, Joël Denis Richard** ◉**, Christopher Leonard Brett** ◉*

Department of Biology, Concordia University, Montreal, Quebec, Canada

* christopher.brett@concordia.ca

## Abstract

Diverse physiology relies on receptor and transporter protein down–regulation and degradation mediated by ESCRTs. Loss–of–function mutations in human ESCRT genes linked to cancers and neurological disorders are thought to block this process. However, when homologous mutations are introduced into model organisms, cells thrive and degradation persists, suggesting other mechanisms compensate. To better understand this secondary process, we studied degradation of transporter (Mup1) or receptor (Ste3) proteins when ESCRT genes (VPS27, VPS36) are deleted in *Saccharomyces cerevisiae* using live-cell imaging and organelle biochemistry. We find that endocytosis remains intact, but internalized proteins aberrantly accumulate on vacuolar lysosome membranes within cells. Here they are sorted for degradation by the intralumenal fragment (ILF) pathway, constitutively or when triggered by substrates, misfolding or TOR activation in vivo and in vitro. Thus, the ILF pathway functions as fail–safe layer of defense when ESCRTs disregard their clients, representing a two–tiered system that ensures degradation of surface polytopic proteins.

## Author summary

Receptor, transporter and channel proteins on the plasma membranes (or surface) of all cells mediate extensive physiology. This requires precise control of their numbers, and damaged copies must be removed to prevent cytotoxicity. Their downregulation and degradation is mediated by lysosomes after endocytosis and entry into the multi–vesicular body (MVB) pathway which depends on ESCRTs (Endosomal Sorting Complexes Required for Transport). Loss-of–function mutations in ESCRT genes are linked to cancers and neurological disease, but cells survive and some proteins continue to be degraded. Herein, we use baker's yeast (*Saccharomyces cerevisiae*) as model to better understand how surface proteins are degraded in cells missing ESCRT genes. Using fluorescence microscopy matched with biochemical and genetic approaches, we find that the methionine transporter Mup1 and G-protein coupled receptor Ste3 continue to be degraded when two ESCRT genes are deleted. They are endocytosed but rerouted to membranes of vacuolar lysosomes after stimuli are applied to trigger their downregulation. Here they are sorted into intralumenal fragments and degraded by acid hydrolases

**Funding:** This work was supported by Discovery Program grant #RGPIN/2017-06652 to C.L.B from the Natural Sciences and Engineering Research Council of Canada (https://www.nserc-crsng.gc. ca). The funders had no role in study design, data collection and analysis, decision to publish, or preparation of the manuscript.

**Competing interests:** The authors have declared that no competing interests exist.

within vacuolar lysosomes upon homotypic membrane fusion. We propose that this intra-lumenal fragment (ILF) pathway functions as a secondary mechanism to degrade surface proteins with the canonical MVB pathway is disrupted.

## Introduction

Precise control of surface receptor, transporter and channel protein lifetimes underlies diverse physiology, including immune responses, endocrine function, tissue development, nutrient absorption, metabolism and synaptic plasticity [1–8]. Damaged surface proteins must also be cleared from the plasma membrane to prevent proteotoxicity [9–11]. These damaged or unneeded polytopic proteins are first labeled and selectively internalized within the cell by endocytosis [10,12–17]. At endosomes, these proteins are packaged into intralumenal vesicles (ILVs) by ESCRTs (Endosomal Sorting Complexes Required for Transport) [18]. Many rounds of ILV biogenesis forms a multivesicular body (MVB) [19,20]. Once mature, the perimeter membrane of the MVB fuses with the lysosome membrane, exposing protein-laden ILVs to lumenal acid hydrolases for degradation [1,21].

Although this ESCRT-dependent MVB pathway is clearly important for proteostasis, reports of ESCRT-independent down-regulation have emerged [22–27]. These observations challenged the presumption that this fundamental process is exclusive, which led to our recent discovery of an alternative called the ILF pathway [28,29]: In the model organism *S. cerevisiae*, we found that some surface transporter proteins bypass ESCRTs on endosomal membranes after endocytosis, and thus remain on perimeter membranes of mature MVBs. Upon subsequent fusion with vacuolar lysosomes (or vacuoles), these transporters are deposited onto vacuole membranes. Here, they are sorted into a disc encircled by a ring of fusogenic lipid and proteins that forms at the vertex between two docked organelles [30,31]. Upon homotypic vacuole membrane fusion at this vertex ring, apposing protein-laden discs merge forming an intralumenal fragment (ILF) that is exposed to lumenal hydrolases and catabolized [32–34].

Protein degradation by the ESCRT-independent ILF pathway or canonical MVB pathway are triggered by similar stimuli, such as protein misfolding by heat stress for quality control, TOR signaling, or changes in substrate levels [10,14,15,17,28,29]. This suggests they play comparable roles in regulating surface protein lifetimes for proteostasis and physiology. However, the basis of pathway selection by client proteins remains obscure, and it unclear whether these pathways can share client proteins. This is important because some human cancers and neurological disorders are linked to loss-of-function mutations in genes encoding ESCRT subunits that presumably block the MVB pathway [35]. Deleting these genes in model organisms has implicated persistence of residual ESCRT-client protein catabolism [36–39], suggesting that another mechanism is compensating for their loss. Importantly, cargo proteins continue to be endocytosed in mutant cells and accumulate on vacuole membranes where they could be recognized by the ESCRT-independent ILF pathway [40–47]. Thus, herein we test the hypothesis that the ILF pathway degrades ESCRT-client proteins when they are not discarded by the canonical MVB pathway.

## Results

### Mup1 is deposited onto vacuole membranes when unrecognized by ESCRTs

Others have observed aberrant deposition of surface receptor and transporter proteins on vacuole membranes in *S. cerevisiae* cells with mutations that prevent sorting into the ESCRT-

dependent MVB pathway [e.g. 40–47]. These include quintessential ESCRT client proteins Ste3, a G-protein coupled receptor critical for pheromone-mediated mating, and Mup1, a high-affinity methionine permease important for metabolism, proteostasis and survival in response to changes in nutrient availability [15,48]. Deposition on vacuole membranes after endocytosis seems like a logical outcome if they cannot be sorted into ILVs at endosomes, as they will remain on the endosome limiting membrane and upon fusion with a vacuole should decorate the membrane of the product (Fig 1A) [29].

To confirm, we first monitored Mup1-GFP in live yeast cells after methionine addition, which triggers its endocytosis and degradation [10,49,50]. In wild type cells, prior to treatment (0 minutes), Mup1-GFP is mostly found on the plasma membrane (Fig 1B and 1C). 5 minutes after exposure, it accumulates on intracellular puncta representing endosomes. Then, after 30 minutes, Mup1-GFP is predominantly found within the lumen of vacuoles where it is catabolized by lumenal acid hydrolases. Consistent with Mup1-GFP being efficiently packaged into ILVs by ESCRTs, we find that it does not appear on vacuole membranes stained with FM4-64 (Fig 1C). We next deleted VPS36, a key subunit of ESCRT-II [51], and confirmed the knockout by genomic sequencing and phenotyping, *e.g.* mutant growth is sensitive to $MnCl_2$ and steady–state CPY maturation is reduced [52–54] (S1 Fig). When studying these *vps36Δ* cells after methionine addition, we found that Mup1-GFP follows a similar trajectory from the plasma membrane to vacuole lumen (Fig 1B), albeit with two important exceptions: (1) Large Mup1-GFP and FM4-64 –positive puncta appeared adjacent to vacuole membranes, a hallmark phenotype of ESCRT deletion mutants (VPS class E) [55], confirming ILV formation was blocked at endosomes in these cells, and (2) Mup1-GFP aberrantly accumulates on vacuole membranes on route to the lumen (Fig 1B and 1C), which is not a trafficking intermediate of the MVB pathway.

To better visualize this intermediate, we replaced the cytoplasmic GFP tag on Mup1 with pHluorin, a pH-sensitive variant of GFP. When exposed to the acidic environment of the vacuole lumen, pHluorin fluorescence is quenched allowing better detection of Mup1-pHluorin on vacuole membranes [45]. As expected, we observe Mup1-pHluorin clearly decorating vacuole membranes only in *vps36Δ* cells (Fig 1D). Upon closer examination of the population of cells imaged, we confirmed that nearly 50% of *vps36Δ* cells studied presented Mup1-pHluorin on vacuole membranes after methionine addition, whereas no wild type cells showed this phenotype (Fig 1E). Because the MVB pathway does not involve this trafficking intermediate, we propose that when ESCRTs fail to sort endocytosed Mup1-GFP into ILVs at endosomes, it remains on endosome perimeter membranes and, upon MVB-vacuole fusion, is deposited on vacuole membranes [see 21].

## Mup1 degradation triggered by methionine persists in an ESCRT mutant

Although the pathway from the plasma membrane may differ, Mup1-GFP is eventually delivered to the lumen of vacuoles in *vps36Δ* cells after methionine addition (Fig 2A, see Fig 1). This is the normal site of Mup1 proteolysis, but it is unclear if Mup1 degradation persists when ESCRT genes are deleted. To test this, we used western blot analysis to detect GFP cleavage from Mup1-GFP, a conventional indicator of proteolysis [see 50]. As previously reported, Mup1-GFP is rapidly degraded after wild type cells are exposed to methionine (Fig 2B). Importantly, Mup1-GFP degradation after methionine addition persevered in *vps36Δ* cells (Fig 2B). Notably, some Mup1-GFP fusion protein remained within the mutants after 2 hours (Fig 2C), when it was completely cleared in wild type cells, suggesting degradation was less efficient without VPS36 as expected.

To demonstrate that vacuole proteases mediate Mup1-GFP cleavage, we treated cells with bafilomycin A1, an inhibitor of the V-type $H^+$-ATPase (V-ATPase), to prevent lumenal

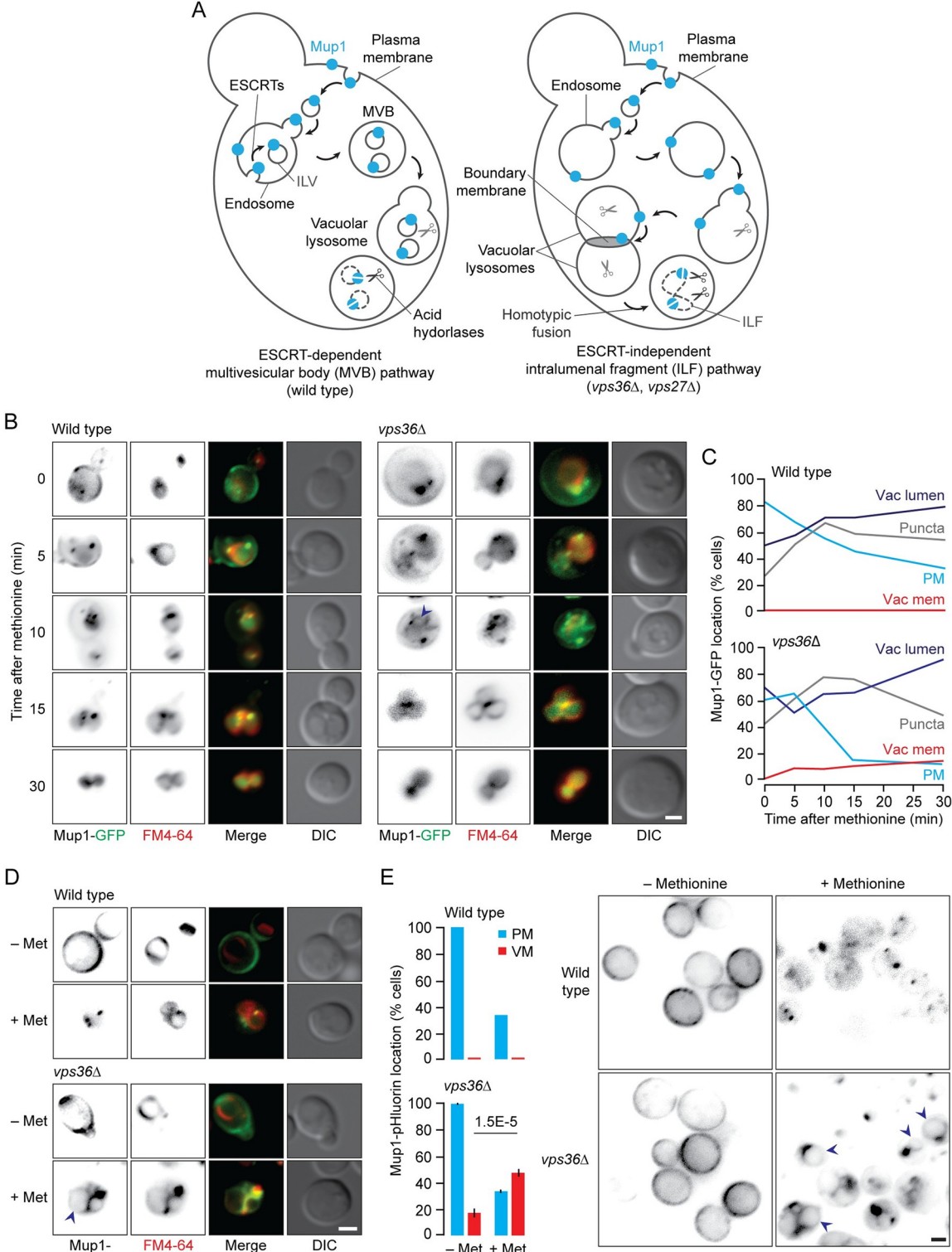

**Fig 1. Mup1 accumulates on vacuole membranes in *vps36Δ* cells after addition of methionine.** (A) Illustration describing how surface ESCRT-client proteins like Mup1 may be rerouted to the vacuole membrane and ILF pathway for degradation when MVB formation is blocked. (B) Fluorescence and DIC micrographs showing route taken by Mup1-GFP from the plasma membrane to the vacuole lumen in response to methionine over 30 minutes in live wild type or *vps36Δ* cells stained with FM4-64. (C) Proportion of wild type or *vps36Δ* cells that show Mup1-GFP fluorescence on the plasma membrane (PM), intracellular puncta, vacuole membrane (Vac

mem) or vacuole lumen over time after methionine addition. n values for wild type (0, 5, 10, 20, 30 min) are 242, 327, 229, 267, 266 cells; *vps36Δ* are 216, 216, 214, 196, 222 cells. (D) Fluorescence and DIC micrographs of live wild type or *vps36Δ* cells stained with FM4-64 expressing Mup1-pHluorin before or 10 minutes after addition of methionine. (E) Proportion of wild type or *vps36Δ* cells that show Mup1-pHluorin fluorescence on the plasma membrane (PM) or vacuole membranes (VM) before or 10 minutes after methionine addition. Micrographs show Mup1-pHluorin location within cells. n values for wild type are 318, 304 cells; *vps36Δ* are 324, 348 cells analyzed before or after methionine addition respectively. Means ± S.E.M. and results from Student t-test are shown. Cells were stained with FM4-64 to label vacuole membranes. Arrowheads indicate Mup1-GFP or Mup1-pHluorin on vacuole membranes. Scale bars, 2 μm.

acidification, a requisite for their activity [56]. Western blot analysis of whole cell lysates revealed that pretreatment with bafilomycin A1 blocked Mup1-GFP cleavage in wild type and *vps36Δ* cells (Fig 2D and 2E). Blocking V-ATPase function may also inhibit membrane fusion between endocytic organelles preventing delivery of Mup1-GFP to the vacuole lumen, accounting for observed effects of bafilomycin A1 on proteolysis. To eliminate this possibility, we visualized live cells using fluorescence microscopy and confirmed that Mup1-GFP is present within the lumen of vacuoles after bafilomycin A1 addition (Fig 2F). Together, these results show that Mup1-GFP continues to be degraded by vacuoles within cells missing VPS36 by a secondary, fail-safe mechanism.

## The ILF pathway mediates Mup1 down-regulation by methionine when ESCRTs are inactivated

Polytopic proteins that reside on vacuole membranes can be selectively degraded by three mechanisms: the VRED (vacuole membrane protein recycling and degradation) pathway, microautophagy, or the ILF pathway [28,57,58]. The presence of internalized Mup1-GFP on vacuole membranes suggests that at least one of these mechanisms is responsible for their degradation. However, the VRED and microautophagic pathways are ESCRT-dependent, eliminating the possibility that they contribute. Moreover, other surface transporters known to be degraded by the ESCRT–independent ILF pathway (e.g. Hxt3, a glucose transporter) appear on vacuole membranes after they are internalized by endocytosis [29]. Thus, we hypothesized that the ESCRT–client Mup1-GFP is degraded by the ILF pathway in *vps36Δ* cells (see Fig 3A).

To be degraded by the ILF pathway, transporter proteins are sorted into an area of the vacuole membrane surrounded by a ring of fusogenic lipids and proteins assembled at the interface or "boundary" between apposing organelles [28,30,31]. We assessed Mup1-pHluorin sorting by measuring pHluorin fluorescence intensity within boundary membranes relative to its intensity in the membrane outside of this area using micrographs of *vps36Δ* cells treated with methionine. We found that Mup1-pHluorin was present within boundary membranes formed between docked vacuoles (Fig 3B) within the population of cells studied (Fig 3C). To better assess Mup1-pHluorin sorting into the ILF pathway, we compared it to GFP-tagged Fet5, a vacuolar copper-iron oxidase that is typically excluded from, or Fth1, a vacuolar iron transporter that is typically enriched in boundary membranes [see 28]. We found Mup1-pHlurorin levels were higher than Fet5-GFP but lower than Fth1-GFP (measured in either *vps36Δ* or wild types cells), suggesting that its sorting into boundary membranes is neither restricted or enhanced (Fig 3B and 3C), and like Fth1 and Fet5, Mup1 sorting into boundaries does not require ESCRTs [28]. Nevertheless, the presence of Mup1-GFP in boundaries suggests it is degraded by the ILF pathway in cells lacking VPS36.

To internalize proteins into the lumen, docked vacuoles undergo homotypic lipid bilayer merger at the vertex ring that surrounds apposing boundary membranes to produce an ILF within the fusion product [28,32]. To confirm that Mup1 is delivered to the vacuole lumen by this mechanism, we recorded homotypic vacuole fusion events in live *vps36Δ* cells exposed to

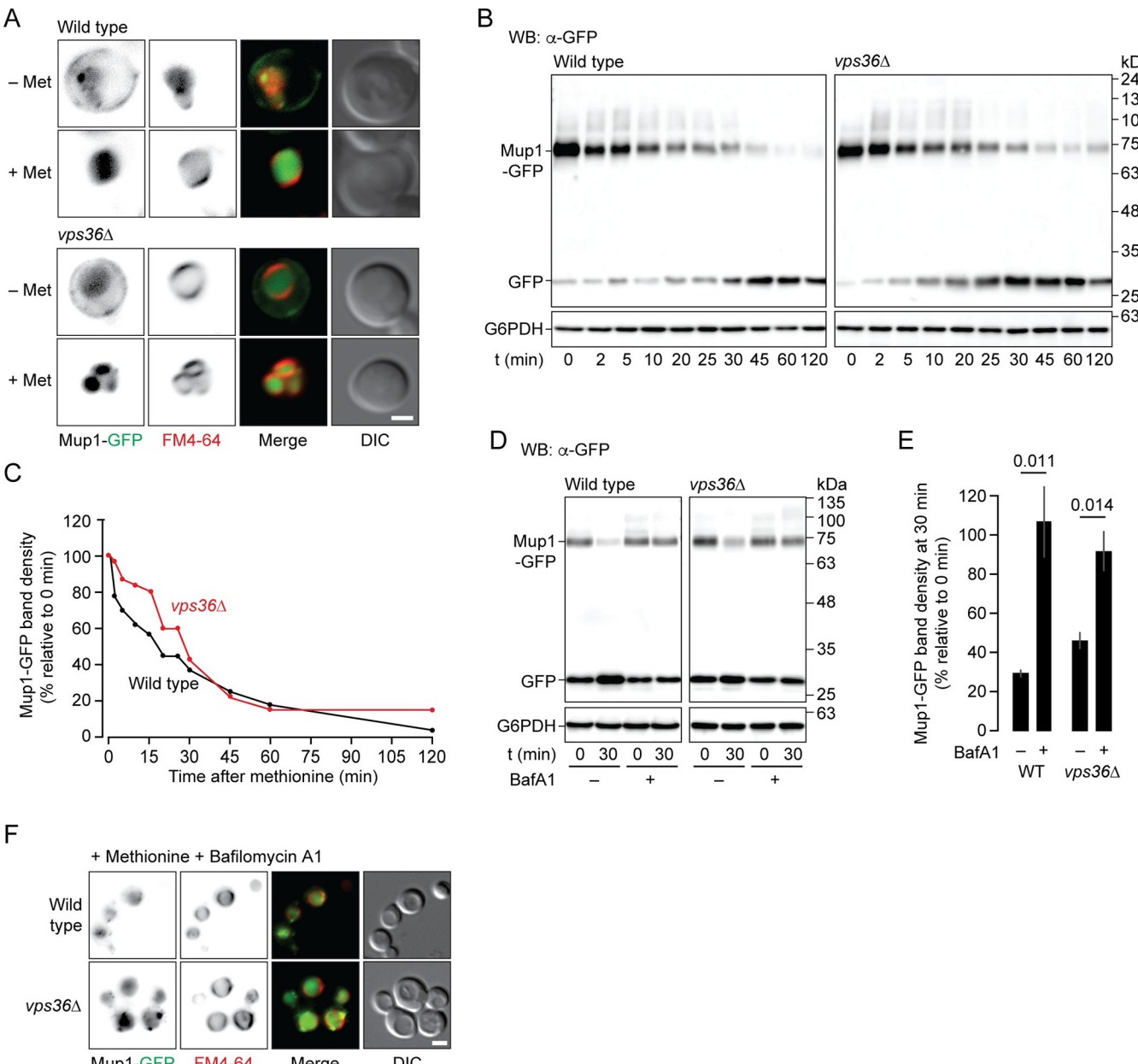

**Fig 2. Mup1 is degraded by vacuoles in *vps36Δ* cells.** (A) Fluorescence and DIC micrographs of live wild type or *vps36Δ* cells stained with FM4-64 and expressing Mup1-GFP before or 30 minutes after addition of methionine. (B) Western blot analysis of whole cell lysates prepared from wild type (WT) or *vps36Δ* cells expressing Mup1-GFP before (0 minutes) or 5–120 minutes after addition of methionine. Blots were stained for GFP or glucose-6-phosphate dehydrogenase (G6PDH; as load controls). Estimated molecular weights shown. (C) Intact Mup1-GFP band densities relative to 0 minutes and normalized to corresponding G6PDH densities were calculated for each time shown in B. n = 3 for each strain tested. (D) Western blot analysis of whole cell lysates prepared from wild type (WT) or *vps36Δ* cells expressing Mup1-GFP before (0 minutes) or 30 minutes after addition of methionine in the presence or absence of 10 μM bafilomycin A1 (BafA1). Blots were stained for GFP or glucose-6-phosphate dehydrogenase (G6PDH; as load controls). Estimated molecular weights shown. (E) Intact Mup1-GFP band densities at 30 minutes relative to 0 minutes and normalized to corresponding G6PDH densities were calculated for each time shown in D. n = 3 for each strain tested. (F) Fluorescence and DIC micrographs of live wild type or *vps36Δ* cells stained with FM4-64 and expressing Mup1-GFP 30 minutes after addition of methionine in the presence of bafilomycin A1. Means ± S.E.M. and results from Student t-test are shown. Cells were stained with FM4-64 to label vacuole membranes. Arrowheads indicate Mup1-GFP or Mup1-pHluorin on vacuole membranes. Scale bars, 2 μm.

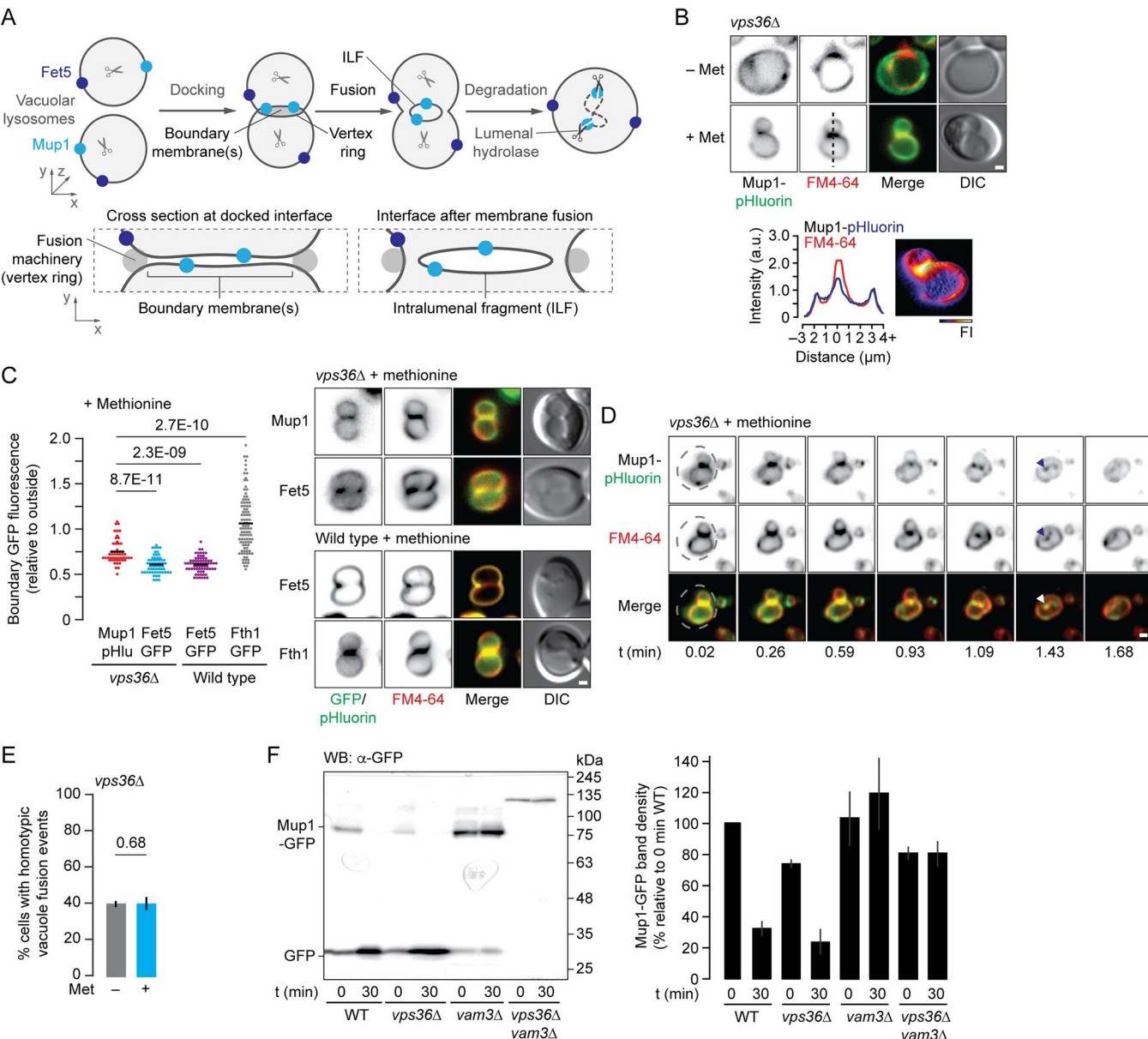

**Fig 3. Methionine triggers Mup1 degradation by the ILF pathway in *vps36Δ* cells.** (A) Cartoon illustrating how Mup1 on vacuole membranes may be sorted into boundaries and ILFs formed during homotypic vacuole fusion. Whereas Fet5 is depleted from boundaries and ILFs, and exclusively resides on outside membranes. (B) Fluorescence and DIC micrographs of live *vps36Δ* cells stained with FM4-64 expressing pHluorin-tagged Mup1 before or 10 minutes after addition of methionine. A 3-dimensional Mup1-pHluorin fluorescence intensity (FI) plot and line plots of pHluorin or FM4-64 fluorescence intensity (line shown in above micrograph) indicate boundary membrane localization after methionine addition. (C) Mup1-pHluorin, Fet5-GFP or Fth1-GFP fluorescence measured within boundary membranes of docked vacuoles within live wild type or *vps36Δ* cells in the presence of methionine. 52 (Mup1-pHluroin, *vps36Δ*), 82 (Fet5-GFP, *vps36Δ*), 70 (Fet5-GFP, wild type), 97 (Fth1-GFP, wild type) boundaries were analyzed. Mup1-pHluorin was absent from vacuole membranes without methionine and thus was not analyzed. (D) Snapshots from time-lapse movie showing a homotypic vacuole fusion event within a live *vps36Δ* cell expressing Mup1-pHluorin stained with FM4-64 10 minutes after methionine addition. Dotted lines indicate cell perimeter; arrowheads indicate newly formed ILF. See S1 Video. (E) Analysis of data shown in *D* indicating proportion of *vps36Δ* cells that displayed a vacuole fusion event within 5 minutes before (n = 1,446) or 10 minutes after methionine addition (1,057). (F) Western blot analysis (left) of whole cell lysates prepared from wild type (WT), *vps36Δ*, *vam3Δ*, or *vps36Δvam3Δ* cells expressing Mup1-GFP before (0 minutes) or 30 minutes after addition of methionine. Estimated molecular weights shown. Intact Mup1-GFP band densities relative to 0 minutes WT were calculated for each time shown (right). n = 3 for each strain tested. Means (bars) ± S.E.M. and results from Student t-test are shown. Vacuole membranes were stained with FM4-64. Scale bars, 1 μm.

methionine using HILO fluorescence microscopy. As expected, Mup1-pHluorin decorating boundary membranes is internalized within an ILF upon homotypic vacuole fusion (Fig 3D and S1 Video). This is consistent with results from western blot analysis (Fig 2B and 2C) and delivery to the vacuole lumen (Fig 1). Notably, methionine addition does not affect frequency of homotypic vacuole fusion events (Fig 3E), suggesting that stimulation of vacuole fusion itself is not required to accommodate Mup1 degradation.

To further implicate membrane fusion (a requisite for ILF formation) in Mup1 degradation, we deleted VAM3 which encodes the Qa-SNARE required for vacuole fusion [21]. If the ILF pathway contributes, we hypothesize that its loss will inhibit Mup1 degradation in *vps36Δvam3Δ* cells. As predicted, cleavage of GFP from Mup1 was abolished in *vps36Δvam3Δ* cells by Western blot analysis (Fig 3F). Of note, Mup1-GFP from double mutant cell lysates migrated slower, consistent with severe protein trafficking defects. Cells missing only VAM3 also showed diminished cleavage. Together, these results suggest that vacuole membrane fusion is important for Mup1 degradation particularly in the absence of ESCRTs. Thus, we conclude that methionine addition triggers Mup1 sorting into the ILF pathway for degradation when ESCRTs are inactive.

### ILF pathway degrades Mup1 in *vps36Δ* or vps27Δ cells for proteostasis

TOR (Target Of Rapamycin) signaling is a critical mediator of cellular proteostasis: when external amino acids are abundant TOR kinase is activated to replace high-affinity transporters (e.g. Mup1), critical for scavenging amino acids when cells are starved, with low-affinity transporters to optimize nutrient uptake [59,60]. Thus, in response to adding methionine to methionine–starved cells, TOR signaling is activated to ubiquitylate Mup1 on the plasma membrane triggering endocytosis and degradation by the MVB pathway [14]. External amino acid sensing can be bypassed by directly activating TOR signaling with cycloheximide to drive Mup1 ESCRT-mediated down–regulation [14,49,61]. But it is unclear if TOR signaling continues to mediate Mup1 degradation by the ILF pathway in *vps36Δ* cells.

Thus, we used HILO fluorescence microscopy to test if cycloheximide continued to induce Mup1 down–regulation in cells missing ESCRT genes. We first confirmed that cycloheximide triggers Mup1-pHluorin endocytosis in wild type cells (Fig 4A). When the MVB pathway is inactivated by deleting VPS36, Mup1-pHluorin endocytosis persists but it aberrantly accumulates on vacuole membranes (Fig 4A and 4B). Cycloheximide also stimulated delivery of Mup1-GFP to vacuole membranes in cells lacking VPS27, a component of ESCRT-0 [62], confirming that mutations targeting different ESCRT complexes show similar phenotypes (Fig 4C). Here, Mup1-GFP is present in boundaries formed between docked organelles (Fig 4C and 4D) that are internalized upon fusion (Fig 4E and S2 Video) and degraded (Fig 4F and 4G). Thus, activation of TOR signaling with cycloheximide triggers Mup1 clearance by the ILF pathway when ESCRTs are inactivated. This result is consistent with the idea that Mup1 ubiquitylation stimulated by TOR kinase is critical for endocytosis of surface Mup1, but is dispensable for directing Mup1 to ESCRTs after delivery to endosomes, as in their absence, Mup1 continues to be degraded by the ILF pathway.

Protein quality control is also critical for cellular proteostasis: Misfolded or damaged polytopic proteins that cannot be refolded by chaperones are cleared from the plasma membrane by the MVB pathway to prevent dysfunction and aggregation that may lead to cell death, and to replenish free amino acid pools for protein biosynthesis [17]. Given that the ILF pathway is also responsible for degrading misfolded polytopic proteins [28,29], we hypothesized that it may mediate Mup1 protein quality control in cells lacking ESCRT activity. To trigger protein misfolding, we subjected cells to heat stress (42˚C for 30 minutes). After confirming

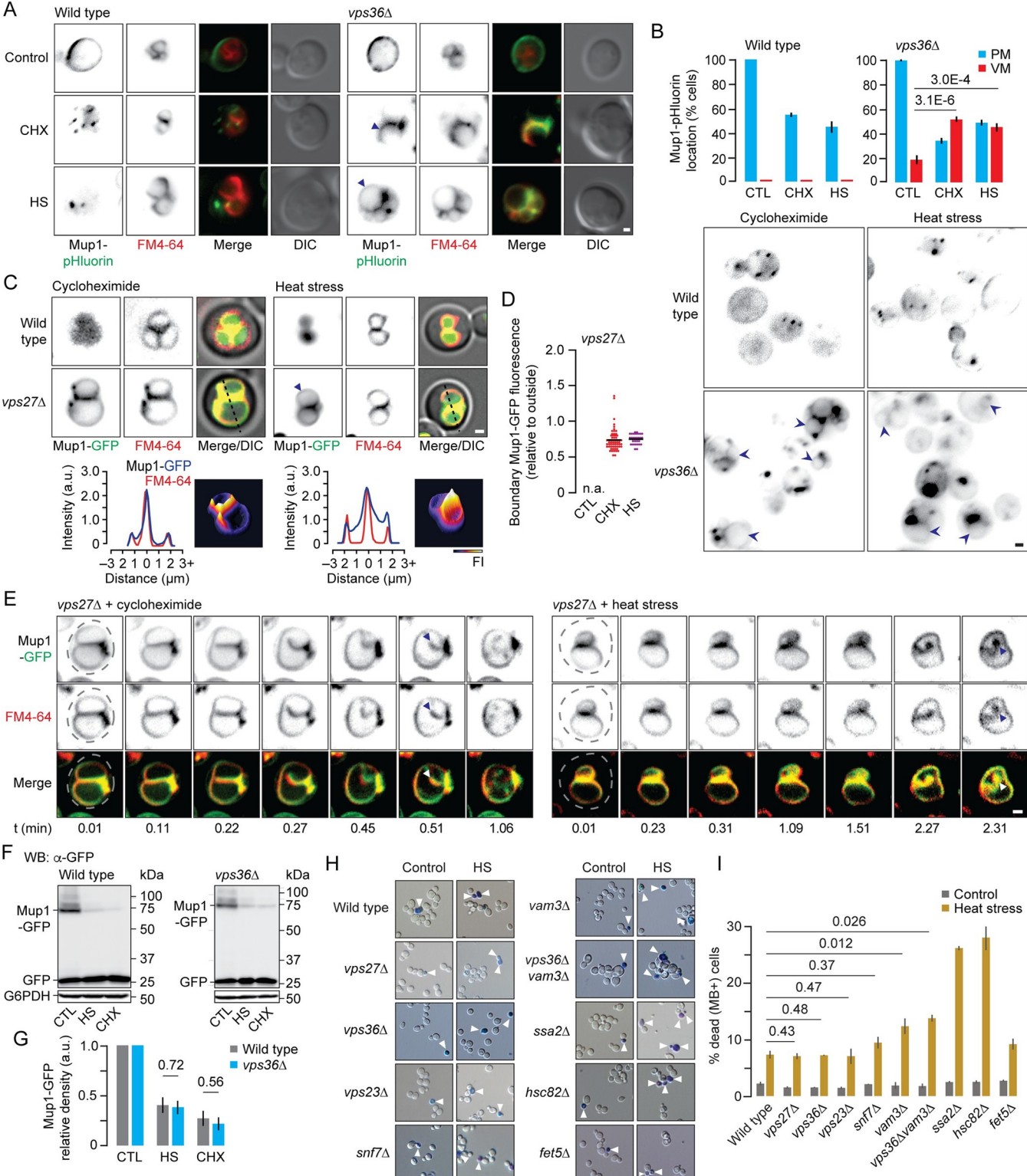

**Fig 4. The ILF pathway degrades Mup1 in response to heat stress or cycloheximide in cells missing ESCRTs.** (A) Fluorescence and DIC micrographs of live wild type or *vps36Δ* cells stained with FM4-64 expressing Mup1-pHluorin before (control) or after addition of 100 μM cycloheximide (CHX) or heat stress (HS; 42°C for 15 minutes). Arrowheads indicate Mup1-pHluorin on vacuole membranes. (B) Proportion of wild type or *vps36Δ* cells showing Mup1-pHluorin fluorescence on the plasma membrane (PM) or vacuole membranes (VM) before or after treatment with cycloheximide (CHX) or heat stress (HS). (below) Micrographs show Mup1-pHluorin location within cells; arrowheads indicate vacuole membranes. Number of cells analyzed (control, HS, CHX) are: WT (318,

315, 365), *vps36Δ* (324,327,339). (C) Fluorescence and DIC micrographs of live wild type or *vps27Δ* cells stained with FM4-64 expressing Mup1-GFP after addition of cycloheximide or heat stress. Three-dimensional Mup1-GFP fluorescence intensity (FI) plots and line plots of GFP or FM4-64 for lines shown above in micrographs indicate boundary membrane localization. (D) Mup1-GFP fluorescence measured within boundary membranes between docked vacuoles within live *vps27Δ* cells before (control; CTL) or after addition of cycloheximide (CHX) or heat stress (HS). Mup1-GFP was absent from vacuole membranes under control conditions and not analyzed (n.a.). Number of *vps27Δ* cells analyzed (HS, CHX) are: 60, 28. (E) Snapshots from time-lapse movies showing homotypic vacuole fusion events within live *vps27Δ* cells stained with FM4-64 expressing Mup1-GFP treated with cycloheximide or heat stress. Dotted lines indicate cell perimeters; arrowheads indicate newly formed ILFs. See S2 and S3 Videos. (F) Western blot analysis of whole cell lysates prepared from wild type or *vps36Δ* cells expressing Mup1-GFP before (control; CTL) or after heat stress (HS) or cycloheximide (CHX) treatment. Blots are stained with anti-GFP or anti-G6PDH antibodies. Estimated molecular weights are shown. (G) Intact Mup1-GFP band densities relative to control and normalized to load controls (G6PDH) were calculated for each condition shown in F. n = 3 for each strain tested. (H) Light micrographs showing methylene blue (MB) stained cultures of wild type, *vps27Δ*, *vps36Δ*, *vps23Δ*, *snf7Δ*, *vam3Δ*, *vps36Δvam3Δ*, *ssa2Δ*, *hsc82Δ* or *fet5Δ* cells before or after heat stress (HS). Arrowheads indicate MB-positive cells. (I) Images in H were used to measure proportion of dead, MB-positive cells in the population. Number of cells analyzed (control, HS) are: 3,232, 1,995 wild type; 2,023, 1,647 *vps27Δ*; 2,226, 1,548 *vps36Δ*; 1,877, 1,743 *vps23Δ*; 1,563, 1,470 *snf7Δ*; 3,021, 1,971 *vam3Δ*; 2,081, 1,540 *vps36Δvam3Δ*; 3,305, 2,017 *ssa2Δ*; 3,401, 2,001 *hsc82Δ*; 2,061, 1,323 *fet5Δ*. Means (bars) ± S.E.M. and results from Student t-test are shown. Vacuole membranes were stained with FM4-64. Scale bars, 1 μm (except in H, 5 μm).

Mup1-GFP degradation was stimulated by heat stress in wild type cells, we determined if the ILF pathway clears Mup1 in response to heat stress in *vps36Δ* or *vps27Δ* cells. As predicted, Mup1 is targeted to vacuole membranes and the ILF pathway for degradation after heat stress when ESCRTs are inactivated (Fig 4A– 4G and S3 Video). Together, these results suggest that the ILF pathway compensates for loss of ESCRTs to ensure protein quality control and cellular proteostasis.

Given that misfolded proteins continue to be cleared by the ILF pathway in the absence of ESCRTs, we hypothesized that this compensatory mechanism should prevent their toxic accumulation and aggregation, ultimately leading to cell death [17]. If true, then deleting ESCRT genes should have no effect on cell survival after heat stress. To test this hypothesis, we measured cell viability by staining yeast cultures with methylene blue to detect dead cells before or after treatment with toxic heat stress (50°C for 30 minutes). As expected, deleting VPS36 (ESCRT-II), VPS27 (ESCRT-0) or other ESCRT genes (VSP23, ESCRT-I; SNF7, ESCRT-III) had no effect on cell viability in response to heat stress as compared to wild type cells (Fig 4H and 4I). As positive controls, we found that heat stress killed most cells lacking genes encoding important protein chaperones activated by heat (SSA2 or HSC82, orthologs of Hsp70 and Hsp90 respectively) [63,64]. As a negative control, deleting FET5, which encodes a copper-iron oxidase with no known role in the canonical heat shock response, has no effect on cell survival after lethal heat stress. Cells lacking VAM3, a Qa-SNARE needed for vacuole fusion and ILF formation, showed sensitivity to heat stress, and cells missing genes needed for both pathways (*vps36Δvam3Δ*) showed greater sensitivity to heat stress (Fig 4H and 4I), suggesting that Vam3-mediated fusion is likely needed to degrade toxic misfolded proteins in cells, especially those lacking ESCRT function. All things considered, we conclude that the ILF pathway compensates for the loss of the MVB pathway by mediating surface amino acid transporter protein down-regulation for cellular proteostasis necessary for survival.

## Mup1-GFP degradation requires ILF pathway machinery when ESCRTs are absent

Polytopic protein sorting into the ILF pathway relies on molecular machinery that co-purifies with vacuoles, in a preparation devoid of cytoplasmic factors that may contribute to Mup1 degradation (e.g. the proteasome) or biosynthesis (ribosomes, endoplasmic reticulum) [28,29]. Thus, to confirm that ESCRT-clients delivered to vacuole membranes uses the ILF machinery for degradation, we isolated vacuoles expressing Mup1–GFP from wild type cells or cells lacking VPS27 and stimulated homotypic vacuole fusion in vitro (by adding physiological salts and ATP) [65]. We imaged reactions using HILO microscopy and first confirmed that no

Mup1-GFP was observed on FM4-64-stained membranes of vacuoles isolated from wild type cells. Rather diffuse GFP fluorescence was exclusively detected in the lumen (as expected; Fig 5A) and this signal was constant over time suggesting no additional Mup1-GFP was

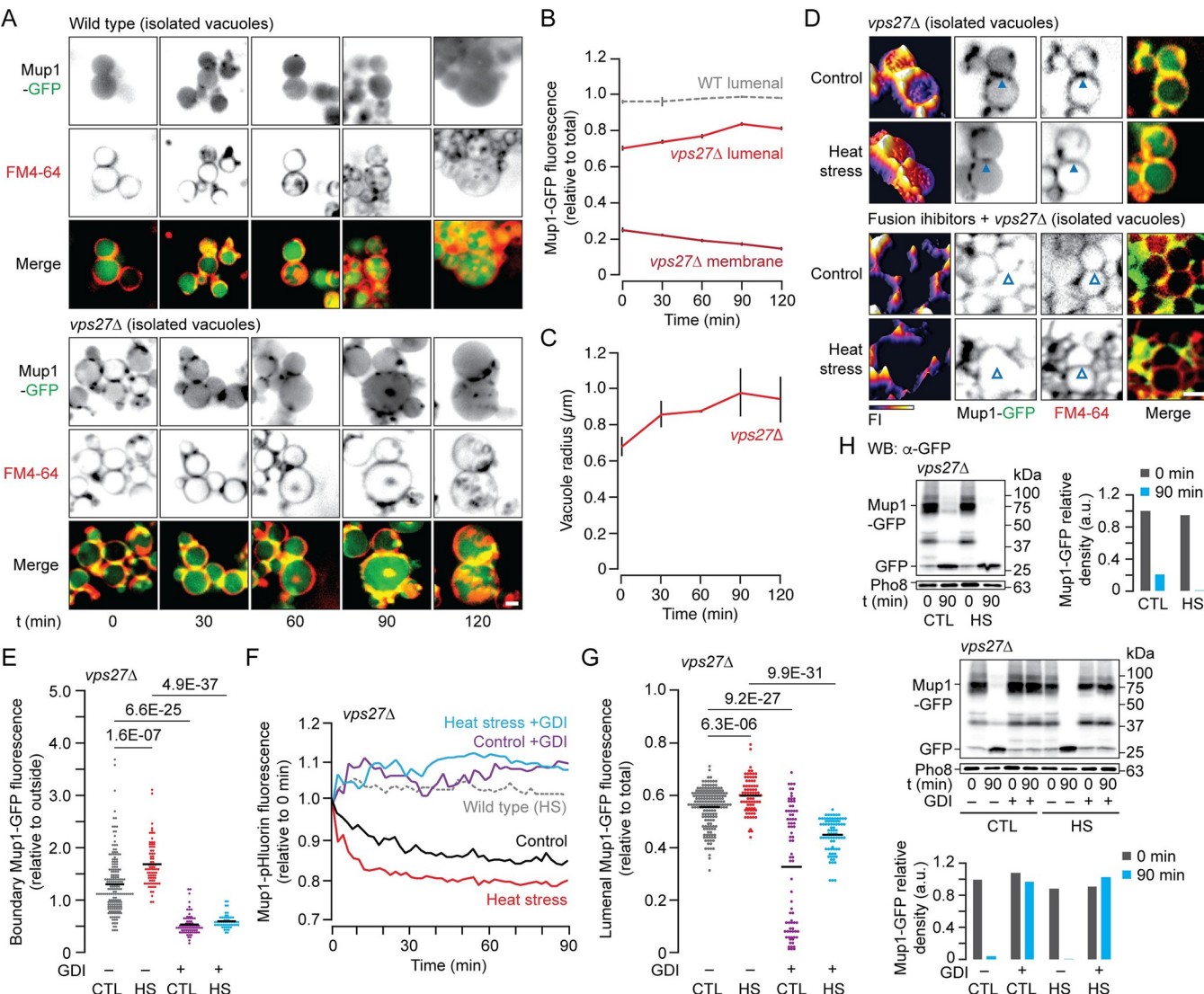

**Fig 5. Molecular machinery for Mup1 degradation by the ILF pathway co-purifies with vacuoles.** (A) Fluorescence micrographs of vacuoles isolated from wild type or *vps27Δ* cells expressing Mup1-GFP before (0 minutes) or after up to 120 minutes of fusion. Vacuole membranes were stained with FM4-64. (B,C) Vacuole radius (C) or Mup1-GFP fluorescence measured within the lumen or on outside membranes (B) of vacuoles isolated from wild type (WT) or *vps27Δ* cells before (0) or after fusion (up to 120 minutes). n values for 0, 30, 60, 90,120 minutes are 63, 52, 83, 115, 114 (WT); 93, 177, 128, 126, 104 (*vps27Δ*) vacuoles analyzed. (D) Fluorescence micrographs of vacuoles isolated from *vps27Δ* cells expressing Mup1-GFP after 30 minutes of fusion in the absence (control) or presence of heat stress (HS), and with or without fusion inhibitors (4 μM rGdi1 and 3.2 μM rGyp1-46). Vacuole membranes were stained with FM4-64 and 3-dimensional fluorescence intensity (FI) plots of Mup1-GFP are shown. Arrowheads indicate boundary membranes either enriched with (closed) or depleted of (open) Mup1-GFP. (E, G) Mup1-GFP fluorescence measured within boundary membranes (E) or the lumen (G) of vacuoles isolated from *vps27Δ* cells in the absence (control; CTL) or presence of heat stress (HS), and with or without fusion inhibitors (GDI) after 30 minutes of fusion. n values for CTL, HS are 176, 76 vacuoles analyzed without GDI; 73, 52 with GDI. (F) Relative fluorescence of vacuoles isolated from WT or *vps27Δ* cells expressing Mup1-pHluorin recorded over time during homotypic vacuole fusion in vitro in the absence (control; CTL) or presence of heat stress (HS) and with or without fusion inhibitors (GDI). n = 3 for each condition, representative traces shown. (H) Western blot analysis of Mup1-GFP degradation before (0) or after (90 minutes) fusion of vacuoles isolated from *vps27Δ* cells in the absence or presence of heat stress (HS) pretreated without (top) or with (bottom) fusion inhibitors (GDI). Blots were stained with antibodies to GFP or Pho8 (load control). Estimated molecular weights indicated. Intact Mup1-GFP band densities relative to control (0 min) normalized to load controls were calculated for each condition shown. n = 3 for each condition tested. Means (bars) ± S.E.M. and results of Student t-tests are shown. Scale bars, 1 μm.

internalized during the fusion reaction in vitro (Fig 5B). However, in vacuoles freshly isolated from *vps27Δ* cells, we found that relatively less Mup1-GFP was in the lumen, which correlated with its presence on outside membranes (Fig 5A). Importantly, outside membrane GFP decreased, whereas lumenal GFP and vacuole size increased over time (Fig 5B and 5C) consistent with Mup1-GFP internalization upon homotypic fusion of vacuoles isolated from *vps27Δ* cells triggered by ATP in vitro. Also, FM4-64-postive vesicles accumulated in the lumen of vacuoles over time, likely representing ILFs formed during homotypic vacuole fusion in vitro (Fig 5A). Notably, Mup1-GFP only appears on ILFs within vacuoles isolated from *vps27Δ* cells over time. These results confirm that selective delivery of Mup1-GFP from the vacuole membrane into its lumen is ESCRT-independent and the underlying machinery co–purifies with the organelles.

Heat stress triggers endocytosis of surface transporter proteins required for delivery to vacuoles for degradation (Fig 4). However, it is unclear whether signaling mechanism triggered by these stimuli also directly facilitate client protein recognition on intracellular membranes by ESCRTs. Whereas, we previously showed that the ILF machinery on isolated vacuole membranes can directly respond to multiple stressors to promote protein degradation [28,29,34]. This is consistent with the vacuole membrane being a major site for TOR signaling protein complex recruitment and assembly for activation [60], and key ubiquitylation machinery and molecular chaperones are known to reside on vacuole membranes [66,67]. Thus, we hypothesized that the sorting and degradation machinery responsible for Mup1-GFP degradation may be directly stimulated by protein misfolding (heat stress). To test this hypothesis, we isolated vacuoles from unstimulated cells, treated them with heat stress under fusogenic conditions, and determined if Mup1-GFP degradation by the ILF pathway was enhanced in vitro.

In support, we found that more Mup1-GFP was enriched in boundary membranes between docked vacuoles isolated from *vps27Δ* cells after heat stress applied in vitro (Fig 5D and 5E), suggesting that protein sorting is stimulated at the vacuole membrane. Next, to determine if internalization was enhanced, we monitored Mup1-pHluorin fluorescence over the course of the fusion reaction [28]. Internalization was observed as a decrease in fluorescence that occurs when pHluorin transitions from the relative high pH within the reaction buffer (pH 6.80) to low pH within the vacuole lumen (pH ~ 5.0). As expected, Mup1-pHluorin fluorescence decreases over time when vacuoles isolated from *vps27Δ* cells undergo fusion in vitro, and treatment with heat stress enhances this effect, particularly the initial rate of internalization (Fig 5F). Notably, pHluorin fluorescence does not change during fusion of vacuoles isolated from wild type cells, confirming our observation that Mup1-pHluorin is exclusively located in the lumen. Consistent with these results, heat stress stimulated Mup1-GFP accumulation in the lumen of vacuoles isolated from *vps27Δ* cells after fusion in vitro (Fig 5G). Moreover, western blot analysis showed cleavage of GFP from Mup1-GFP observed after 90 minutes of vacuole fusion in vitro was enhanced after heat stress (Fig 5H). Together, these results suggest that protein quality control mechanisms known to drive endocytosis of presumably misfolded surface Mup1-GFP may also activate machinery on vacuole membranes to optimize its degradation when ESCRTs are dysfunctional.

Activation of the resident vacuolar Rab-GTPase Ypt7 is implicated in sorting proteins into the ILF pathway prior to membrane fusion [28]. Thus, if Mup1-GFP is down–regulated by the ILF machinery then inhibition of Ypt7 should block its degradation. To test this hypothesis, we acutely blocked Ypt7 by adding two purified recombinant protein inhibitors to vacuoles isolated from *vps27Δ* cells: rGdi1, which extracts Ypt7 from membranes rendering it inactive, and rGyp1-46, the catalytic domain of the Rab-GTPase activating protein Gyp1 that inactivates Ypt7 [68]. As expected, we found that these Ypt7 inhibitors blocked Mup1-GFP sorting into boundary membranes (Fig 5D and 5E), Mup1-pHuorin internalization (Fig 5F), Mup1-GFP

lumenal accumulation (Fig 5G) and its proteolysis (Fig 5H) under standard conditions or after further stimulation by heat stress. These results confirm that Mup1 degradation requires the ILF machinery when ESCRTs are defective.

### Degradation of the G-protein coupled receptor Ste3 is mediated by MVB and ILF pathways

To determine if surface receptor proteins, in addition to transporters, may rely on the ILF pathway for degradation when the MVB pathway is impaired, we examined cells expressing GFP-tagged Ste3, a surface G-protein coupled receptor, and classical ESCRT-client, that is constitutively degraded by the MVB pathway [44,69]. In untreated wild type cells, Ste3-GFP is present on the plasma membrane and within the vacuole lumen, which correlates with a relatively large proportion of cleaved GFP (versus full length Ste3-GFP) observed by western blot analysis (Fig 6A). As predicted, degradation of Ste3-GFP persists in unstimulated *vps27Δ* cells (Fig 6A), where it accumulates on vacuole membranes (Fig 6B and 6C). We made similar observations when treating cells with heat stress to further stimulate Ste3-GFP degradation (Fig 6A– 6C). Like Mup1 in *vps27Δ* cells, Ste3-GFP was present in boundary membranes between docked vacuoles (Fig 6C and 6D) where it was internalized into the lumen upon homotypic vacuole fusion (Fig 6E and 6F and S4 Video). Thus, we conclude that the ILF pathway is capable of mediating down–regulation of the receptor protein Ste3 when unrecognized by ESCRTs.

When we repeated these experiments with pHluorin in place of GFP to better assess the presence of Ste3 on vacuole membranes (without interference of fluorescence signal from the lumen), we discovered that Ste3-pHluorin sometimes appeared on vacuole membranes in wild type cells, whether unstimulated (6% cells examined) or treated with heat stress (16% cells; Fig 6G–6I). This important finding proposes that even in the presence of ESCRTs, the ILF pathway may in part contribute to Ste3 down–regulation in wild-type cells (see Fig 7).

## Discussion

### A new two-tier model for surface receptor and transporter down-regulation

Prior to this study, we discovered that the ILF pathway selectively down–regulates half of surface polytopic proteins studied (3 of 6) in living wild type *S. cerevisiae* cells, including transporters for sugars (Hxt3), myo-inositol (Itr1) and amino acids (Aqr1) [29]. The remainder included Mup1 and Ste3, quintessential ESCRT-clients known to be degraded by the MVB pathway. Deleting ESCRT genes to block the MVB pathway, had no effect on ILF-client protein degradation, confirming that the underlying sorting machinery must be distinct. Thus, these two processes seemed to function independently, whereby each pathway was responsible for down-regulating a unique subset of surface client proteins.

If this existing model is accurate, then ESCRT-client protein down-regulation should be blocked when the MVB pathway is disrupted by genetic mutation. Herein, we tested this hypothesis and provide extensive evidence that it is false: After down-regulation was triggered by multiple stimuli, both ESCRT-client proteins studied (Mup1, Ste3) continued to be delivered to the vacuole lumen and degraded in mutant cells. After endocytosis, they were aberrantly deposited on vacuole membranes, as observed but not discussed in previous reports [e.g. 40–47], likely due to MVB/endosome–vacuole membrane fusion which is less efficient when ESCRTs are deleted [21], but not completely blocked as previously implicated [53]. This location does not represent a trafficking intermediate in the MVB pathway, implying that a

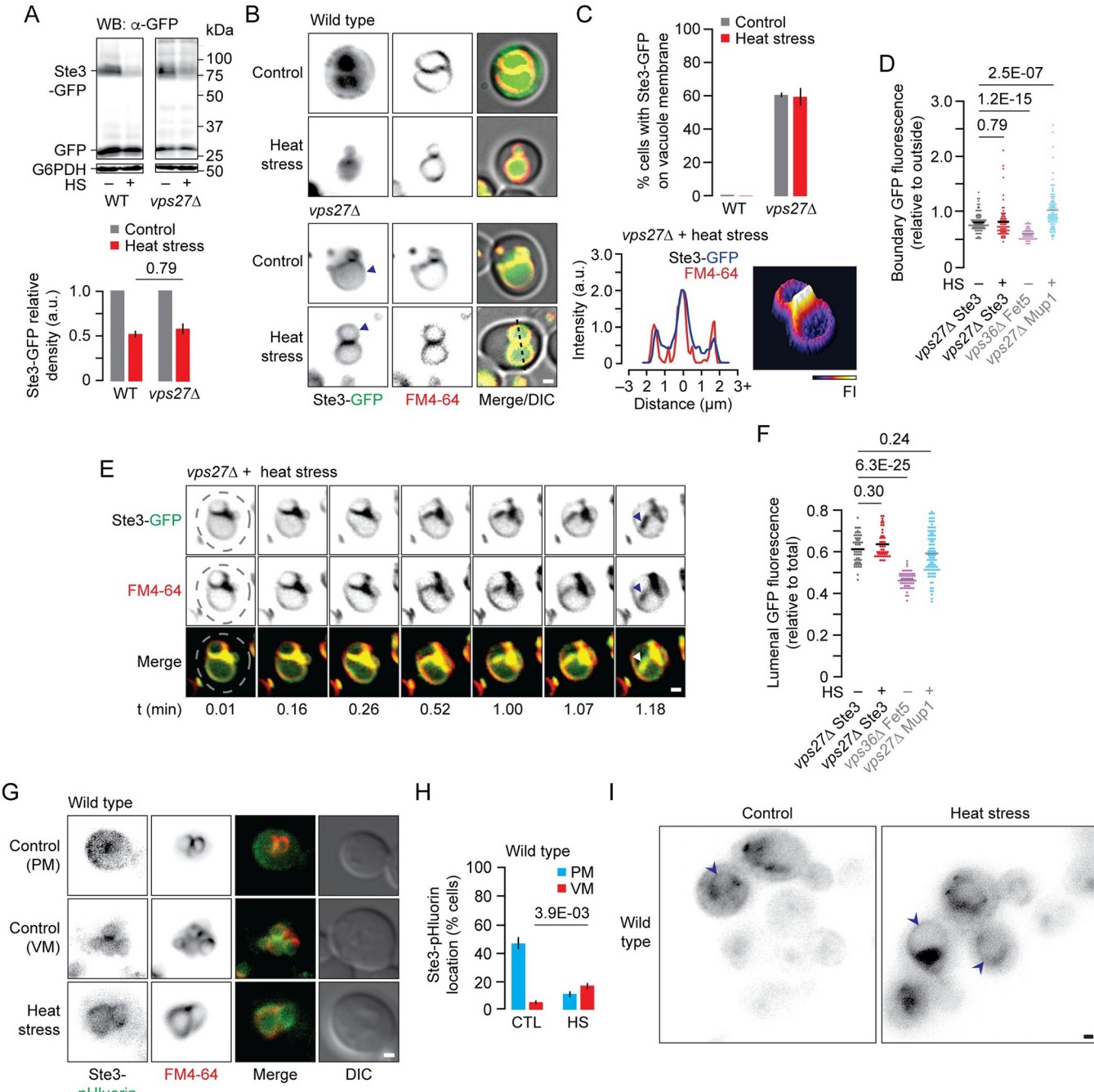

**Fig 6. MVB and ILF pathways mediate degradation of Ste3.** (A) Western blot analysis of whole cell lysates prepared from wild type (WT) or *vps27Δ* cells expressing Ste3-GFP that were untreated or exposed to heat stress (HS). Blots were stained for GFP or G6PDH (load controls). Estimated molecular weights are shown. Intact Ste3-GFP band densities relative to control and normalized to corresponding G6PDH densities were calculated. n = 3 for each strain tested. (B) Fluorescence and DIC micrographs of live wild type or *vps27Δ* cells stained with FM4-64 expressing Ste3-GFP before (control) or after heat stress (HS). Arrowheads indicate vacuole membranes. (C) Proportion of WT or *vps27Δ* cells that show Ste3-GFP fluorescence on vacuole membranes before (control) or after heat stress. 885, 964 WT and 284, 166 *vps27Δ* cells were analyzed before or after heat stress respectively. A 3-dimensional Mup1-GFP fluorescence intensity (FI) plot and line plots of GFP or FM4-64 for line shown in B indicate vacuole and boundary membrane localization. (D,F) Ste3-GFP fluorescence measured within boundary membranes (D) or lumen (F) of vacuoles within live *vps27Δ* cells before (n = 93) or after heat stress (HS; n = 73). For comparison, boundary fluorescence of Fet5-GFP in *vps36Δ* or Mup1-GFP in *vps27Δ* cells after methionine addition are shown (see Fig 2C). (E) Snapshots from a time-lapse movie showing a homotypic vacuole fusion event within a live *vps27Δ* cell stained with FM4-64 expressing Ste3-GFP after heat stress. Dotted line indicates cell perimeter; arrowhead indicates newly formed ILF. See S4 Video. (G) Fluorescence and DIC micrographs of live wild type cells stained with FM4-64 expressing Ste3-pHluorin before (control) or after heat stress. (H) Proportion of wild type cells that show Ste3-pHluorin fluorescence on the plasma membrane (PM) or vacuole membranes (VM) before (control, CTL; n = 195) or after heat stress (HS; n = 198). (I) Micrographs show Ste3-pHluorin location within cells;

arrowheads indicate vacuole membranes. Means (bars) ± S.E.M. and results of Student t-tests are shown. Cells were stained with FM4-64 to label vacuole membranes. Scale bars, 1 μm.

secondary, ESCRT-independent process is responsible their down-regulation. Using many approaches, we demonstrate that these proteins are degraded by the ILF pathway. In particular, we visualized pHluorin–or GFP–tagged proteins being internalized on newly formed intralumenal fragments during homotypic vacuole membrane fusion events in live mutant cells (S1–S4 Videos). This unambiguously shows that ESCRT-client proteins are sorted for degradation by the ILF pathway when they are not sorted into the MVB pathway. Also, the G-protein coupled receptor Ste3 seems to be degraded by the ILF pathway in some wild type cells (Fig 6), suggesting that this secondary pathway may contribute to ESCRT-client protein degradation even when the MVB pathway is intact.

With these new observations in mind, we propose that a two-tier system mediates surface receptor and transporter down–regulation (Fig 7): After endocytosis, proteins first encounter ESCRTs on endosome membranes and can be sorted into ILVs and the MVB pathway. If not detected, they remain on the MVB perimeter membrane and upon heterotypic fusion are deposited on vacuole/lysosome membranes. Here they are cleared by the ILF pathway. This fail–safe mechanism ensures all proteins are degraded.

We speculate that the ILF pathway represents a default mechanism for surface protein down–regulation, because lysosome physiology is reliant on membrane fusion: Lysosomes and vacuoles must undergo heterotypic membrane fusion to receive biomaterials from endocytic

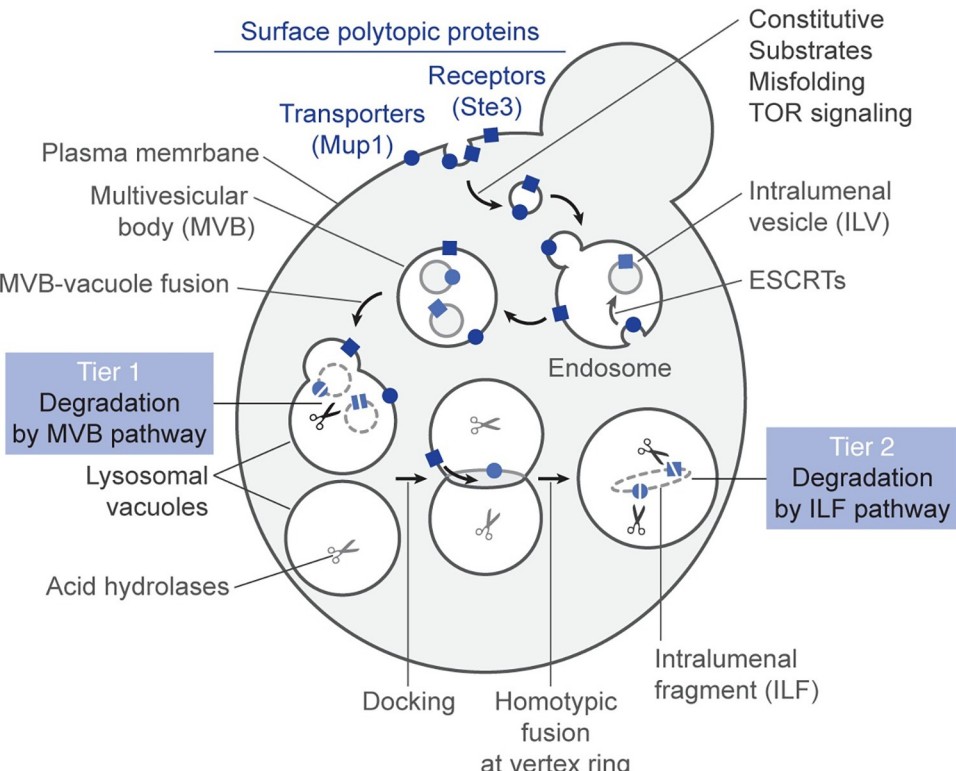

**Fig 7. A two-tiered system ensures surface polytopic protein down-regulation.** Illustration showing how surface transporter and receptor proteins like Mup1 and Ste3, respectively, are degraded by a two-tiered system consisting of the MVB and ILF pathways.

and autophagy pathways [19,21]. They undergo homotypic fusion (and fragmentation) to regulate their morphology–size, shape and number–necessary for organelle inheritance or compartmental expansion for example [56]. The machinery underlying these different fusion events is thought to be nearly identical [21,65,70,71], suggesting that ILFs may be generated during heterotypic as well as homotypic fusion. ILF formation during homotypic lysosome fusion can be regulated: Some fusion events do not produce visible ILFs in unstimulated cells [32–34]. Whereas, in addition to promoting protein sorting into ILFs, stressors increase the probability of ILF formation without changing frequency of fusion events [34]. This also seems to be the case for Mup1 degradation after methionine addition, as this stimulus did not affect frequency of vacuole fusion events (Fig 3E). Thus, every lysosomal fusion event required for organelle function or homeostasis, offers an opportunity to discard unneeded or damaged proteins by sorting them into ILFs. This includes newly deposited surface proteins which are likely recognized as foreign (non-resident, mislocalized) and cleared from lysosome membranes by the ILF pathway to maintain organelle identity.

On the other hand, since their discovery, ESCRTs have been observed at many sites within cells where they contribute to diverse processes besides the MVB pathway, from cytokinesis to plasma membrane repair to extraction of nuclear pore complexes [18,72,73]. Not all ESCRT-client proteins decorating ILVs will necessarily be catabolized, as some MVBs fuse with the plasma membrane, releasing ILVs as extracellular vesicles to mediate intercellular signaling [74,75]. Thus, we speculate that ESCRTs perhaps play a secondary, specialized role in surface protein down–regulation. For example, by acting relatively early in the endocytic pathway, ESCRTs may be limited to recognizing internalized surface client proteins that require immediate sequestration upon endocytosis to quickly terminate signaling (e.g. GPCRs), or to clear proteins that may disrupt function or integrity of endocytic organelles [76]. Although further study is required to better elucidate their distinct roles in proteostasis, it is now clear that both the MVB and ILF pathways are important contributors to surface polytopic protein down–regulation.

### How are client proteins recognized by both pathways?

To be recognized and sorted into the MVB pathway, ESCRT-client proteins are labeled with ubiquitin [16,49,61]. On the other hand, mechanisms responsible for protein labeling and sorting into ILFs have not been elucidated. However, these new results offer insight into this process:

Herein, we show that the ILF pathway degrades surface proteins known to be labeled with ubiquitin, consistent with previous results [29]. Also, the ILF pathway responds to the same stimuli that trigger protein ubiquitylation for entry into the MVB pathway [10,14,15,28,29,61]. For example, in response to amino acids, TOR signaling is activated to dephosphorylate the alpha-arrestin Art1, an adapter protein, which then binds Mup1 to promote ubiquitylation by the HECT E3 ubiquitin ligase Rsp5, a Nedd4 ortholog [50,77–79]. Notably client protein ubiquitylation is required for endocytosis (although contentious) [e.g. 41], and this process does not require ESCRTs [14,80]. In support, we find that client protein endocytosis and upstream signaling mechanisms remain intact in cells missing ESCRTs when they are degraded by the ILF pathway (see Figs 1 – 4 and 6). From this perspective, both pathways seem to rely on protein ubiquitylation at the surface to drive endocytosis, a requisite for delivery to vacuolar lysosomes.

At endosomes, ubiquitylated surface receptor or transporter proteins are added to a complex membrane protein landscape that includes resident polytopic proteins, biosynthetic proteins on route to lysosomes, and proteins marked for degradation from other cell compartments. Here, at this major traffic intersection, functional proteins can be

deubiquitylated and returned to the plasma membrane or their site of origin (i.e. recycled), or remain ubiquitylated to be sorted into ILVs by ESCRTs for degradation. However, this original binary model has since become contentious, as recent data questions the requirement of original ubiquitin labels for protein sorting into ILVs [e.g. 17] or suggests a secondary intracellular labeling event may be required for ESCRT recognition after endocytosis [see 14,16,76]. Our findings also challenge this model by bringing attention to a third option, whereby some internalized surface proteins can be ignored by ESCRTs and retrograde trafficking mechanisms, remain on perimeter membranes of maturing MVBs, and after MVB-vacuole fusion, appear on vacuole membranes. Here, the molecular machinery can directly respond to stimuli to promote client protein degradation by the ILF pathway (Fig 5) [also see 28,29]. Like endosomal membranes, protein labeling machinery is also found on vacuole membranes [57,66,67,81]. This supports the idea that labeling machinery present on endosome or vacuolar lysosome membranes likely contributes to sorting of client proteins into the MVB or ILF pathways for degradation.

What stimulates protein sorting into ILFs on vacuole membranes? One possibility is a proposed mechanism to protect organelle membrane identity: Extracellular faces of receptor or transporter proteins are glycosylated with different sugars and linkages than those covalently attached to lumenal faces of resident vacuole transporter proteins [82]. The surface-specific sugar groups may not offer protection from attack by lumenal hydrolases. Thus, we speculate that, upon delivery to vacuole membranes, surface proteins are particularly susceptible to cleavage, triggering misfolding, which in turn recruit mechanisms on the vacuole membrane to label and sort them into the ILF pathway. Although speculative, this model is supported by the observation that heat stress, to promote misfolding, applied to vacuole fusion reactions further stimulates Mup1 degradation in vitro (Fig 5). How misfolded proteins are then recognized is unclear, but is the focus of ongoing studies important for understanding how ILF and MVB pathways share the burden of surface transporter and receptor down-regulation.

A recent report challenges whether ubiquitylated proteins are degraded by the ILF pathway, arguing that ESCRT–mediated microautophagy is exclusively responsible instead [83]. We do not directly test this hypothesis in our study, nor do we present data suggesting ubiquitin is required for protein degradation by the ILF pathway–although it is the focus of ongoing research. However, our results show that when triggered by methionine, cycloheximide or heat stress, GFP–or pHluorin–tagged Mup1 is degraded within the lumen of vacuoles in cells missing ESCRT genes VPS36 or VPS27 (Figs 2 and 4). Because intact ESCRTs mediate this form of microautophagy [83], it unlikely contributes to Mup1 degradation in these mutant cells. In the near real time videos presented (Figs 3D and 4E and S1–S3 Videos), a vacuole membrane structure resembling intermediates of both pathways are observed within live cells: A lumenally protruding tube or flap connected to the outside membrane on one end and severed on the other side is a structure formed after partial Rab–and SNARE–dependent *fusion* of *two vacuoles* immediately prior to ILF formation [28,33,34]. A similar structure, called a macroautophagic tubule, is formed after invagination of the membrane lining *a single vacuole*, and subsequent membrane *fission* by ESCRT-III generates lumenal vesicles [58,83]. Prior to observing these intermediates in these videos, two tethered vacuoles are present and an intact interface containing Mup1–GFP between them is observed spanning the entire contact site, demonstrating that resulting lumenal structures (intermediates and ILFs) are most likely products of membrane fusion. In support, deleting VAM3 a vacuole Qa-SNARE required for fusion in *vps36Δ* cells blocks Mup1 degradation in vivo. Blocking vacuole fusion in vitro with the Rab inhibitor Gdi1 also inhibits Mup1 degradation, confirming that vacuole fusion is necessary. When considering additional evidence presented using complementary approaches, we conclude that the ILF pathway mediates protein degradation in ESCRT mutants.

## Relevance to physiology and disease

Selective down-regulation of surface receptor, transporter and channel proteins triggered by second messenger signaling or substrates underlies diverse physiology in all eukaryotes. Thousands of genes, more than 30% of eukaryotic genomes, are thought to encode membrane proteins [e.g. 84]. However, the degradation of only a few surface membrane proteins has been studied in molecular detail. In particular, evidence directly implicating ESCRTs and the canonical MVB pathway in this process is scarce. Herein we show that two of the best characterized ESCRT-clients can be degraded by the ILF pathway, either constitutively or to accommodate proteostasis and protein quality control, fundamental processes necessary for cell survival (Figs 1–6). Misfolded proteins that supposedly evade endoplasmic-reticulum-associated degradation (ERAD) can also be delivered to the MVB pathway for clearance (e.g. Slg1) [9]. When this is blocked, they accumulate on vacuole membranes and may be degraded by the ILF pathway, possibly broadening its role in cellular protein quality control. Thus, we speculate that the ILF pathway may play an equally important role in physiology as the canonical MVB pathway, as key components of a two-tiered system for selective lysosomal degradation of polytopic proteins.

Lumenal vesicles that resemble ILFs within lysosomes have been observed in organisms ranging from yeast to plants to humans, since the discovery of this organelle decades ago (also called "dense bodies") [e.g. 85]. It is not currently resolved if they are products of membrane fusion or ESCRTs. However, all known machinery underlying ILF formation is conserved in all eukaryotes [34,70]. Similar to our findings in *S. cerevisiae*, surface receptor proteins accumulate on vacuole membranes when ESCRT genes (e.g. CHMP1) are deleted in plants including PIN1, PIN2 and AUX1 which are critical for auxin signaling underlying development [86]. Thus, it is possible that a similar two-tiered system is at play in plant cells. Loss-of-function mutations in human ESCRT genes are linked to cancers and neurodegenerative disorders, and it was proposed that etiology may involve improper client protein down–regulation [35,87,88]. For example, pathogenesis of cancers could be driven by improper surface integrin protein degradation, preventing termination of oncogenic signaling, as implicated in sporadic lung adenoma [89]. Whereas, ESCRT mutations could promote gradual accumulation and possible aggregation of surface proteins causing neural cell dysfunction and premature cell death, a common mechanism thought to underlie neurodegenerative diseases [90]. Although defects in other ESCRT-mediated cellular processes may contribute to pathology [91], our findings offer a new potential therapeutic strategy to possibly overcome defects in protein down–regulation by targeting the ILF pathway, e.g. through TFEB (Transcription Factor EB) to increase lysosome numbers and/or fusogenicity [92,93].

## Materials and methods

### Yeast strains and reagents

*Saccharomyces cerevisiae* strains used in this study are listed in Table 1. Mat-A yeast deletion clones used are from the complete set purchased from Invitrogen Corp. (Cat# 95401.H2; Carlsbad, USA). Strains generated for this study are available from the corresponding author upon reasonable request. We knocked out VPS27, VPS36 or VAM3 by homologous recombination using the Longtine method [94] and primer sets described previously [29]. Genomic mutations were confirmed by sequencing. GFP or pHluorin genes were integrated into the genome behind genes of interest to ensure minimal effects on native gene expression. All reagents for molecular biology (enzymes, polymerases, ligases) were purchased from New England Biolabs (Ipswich, USA). Biochemical and yeast growth reagents were purchased from

**Table 1.** *Saccharomyces cerevisiae* strains used in this study.

| Strain | Genotype | Source |
|---|---|---|
| SEY6210 | *MATα, leu1-3 ura3-52 his3-200 trp1-901 lys2-801 suc2-D9* | [98] |
| BY4741 | *MATa his3-Δ1 leu2-Δ0 met15-Δ0 ura3-Δ0* | [99] |
| Fet5-GFP | BY4741, *Fet5-GFP::HIS3MX* | [99] |
| Fth1-GFP | BY4741, *Fth1-GFP::HIS3MX* | [99] |
| Fet5-GFP:: *vps36Δ* | BY4741, *Fet5-GFP::HIS3MX, vps36Δ:KanMX* | [28] |
| Mup1-GFP | SEY6210, *Mup1-GFP::KanMX* | [45] |
| Mup1-GFP:*vps27Δ* | SEY6210, *Mup1-GFP::KanMX, vps27Δ:HIS3MX* | This study |
| Mup1-GFP:*vps36Δ* | SEY6210, *Mup1-GFP::KanMX, vps36Δ:HIS3MX* | This study |
| Mup1-pHluorin | SEY6210, *Mup1-pHluorin::KanMX* | [45] |
| Mup1-pHluorin:*vps27Δ* | SEY6210, *Mup1-pHluorin::KanMX, vps27Δ:HIS3MX* | This study |
| Mup1-pHluorin:*vps36Δ* | SEY6210, *Mup1-pHluorin::KanMX, vps36Δ:HIS3MX* | This study |
| Mup1-GFP:*vam3Δ* | SEY6210, *Mup1-GFP::KanMX, vam3Δ:TRP1* | This study |
| Mup1-GFP:*vps36Δvam3Δ* | SEY6210, *Mup1-GFP::KanMX, vps36Δ:HIS3MX, vam3Δ:TRP1* | This study |
| Ste3-GFP | SEY6210, *Ste3-GFP::KanMX* | [45] |
| Ste3-pHluorin | SEY6210, *Ste3-pHluorin::KanMX* | [45] |
| Ste3-GFP:*vps27Δ* | SEY6210, *Ste3-GFP::KanMX, vps27Δ:HIS3MX* | This study |
| *vps27Δ* | BY4741, *vps27Δ:KanMX* | Invitrogen Corp. |
| *vps36Δ* | BY4741, *vps36Δ:KanMX* | Invitrogen Corp. |
| *hsc82Δ* | BY4741, *hsc82Δ:KanMX* | Invitrogen Corp. |
| *ssa2Δ* | BY4741, *ssa2Δ:KanMX* | Invitrogen Corp. |
| *snf7Δ* | BY4741, *snf7Δ:KanMX* | Invitrogen Corp. |
| *vps23Δ* | BY4741, *vps23Δ:KanMX* | Invitrogen Corp. |
| *fet5Δ* | BY4741, *fet5Δ:KanMX* | Invitrogen Corp. |

Sigma-Aldrich (Oakville, Canada), Thermo-Fisher Scientific (Burlington, Canada), or BioShop Canada Inc (Burlington, Canada). Recombinant rabbit IgG raised against GFP (B2) or mouse monoclonal IgG against Pho8 (1D3A10) were purchased from Abcam (Cat# ab290, Cat# ab113688; Toronto, Canada), rabbit IgG against G6PDH from Sigma-Aldrich (Cat# A9521) and mouse IgG against CPY from Invitrogen (Cat# A-6428). Horseradish peroxidase-labeled affinity purified IgG to rabbit or to mouse was purchased from SeraCare (Cat# 5450–0010, 5450–0011; Milford, USA). Bafilomycin A1 was purchased from Sigma-Aldrich (Cat# B1793). Recombinant Gdi1 protein was purified from *E. coli* cells using a calmodulin-binding peptide intein fusion method [68]. Recombinant Gyp1-46 protein (representing the catalytic domain of Gyp1, a Rab-GTPase activating protein) was purified as previously described [95]. Reagents used in vacuole fusion reactions were prepared in 20 mM Pipes-KOH, pH 6.8, and 200 mM sorbitol (Pipes Sorbitol buffer, PS).

## Western blot analysis

For analysis of whole cell lysates, yeast cells were prepared as previously described [96]. Cells were grown in culture to mid-log phase, washed once with YPD (yeast extract peptone dextrose) medium, resuspended in fresh YPD medium, and incubated at 30°C in the presence or absence 100 μM cycloheximide for 120 minutes, or incubated at 42°C for 90 minutes for heat stress. To trigger down-regulation of Mup1 by methionine, cells were instead grown in culture to mid-log phase in washed synthetic complete medium lacking methionine (SC–met), collected and washed once with SC–met, then resuspended in SC with 0.2 mM methionine and incubated at 30°C for up to 120 minutes. To block acid hydrolases, cells were treated with

10 μM bafilomycin A1 for 30 minutes at 30°C prior to adding methionine. After treatment, 5 $OD_{600nm}$ units of cells were collected, resuspended in 0.5 mL of lysis buffer (0.2 M NaOH, 0.2% β-mercaptoethanol) and incubated on ice for 10 minutes. Samples were then treated with trichloroacetic acid (5% final concentration) and incubated on ice for an additional 10 minutes. Precipitates were collected by centrifugation (12,000 x g for 5 minutes at 4°C) and resuspended in 35 μL of dissociation buffer (4% SDS, 0.1 M Tris-HCl, pH 6.8, 4 mM EDTA, 20% glycerol, 2% β-mercaptoethanol and 0.02% bromophenol blue). Samples were treated with Tris-HCl, pH 6.8 (0.3 M final concentration) and incubated at 37°C for 10 minutes. For analysis of in vitro fusion reactions, samples were prepared from isolated vacuoles as previously described [28].

Whole cell lysates or isolated vacuole preparations were then loaded into 10% SDS-polyacrylamide gels. After electrophoresis, separated proteins were transferred to nitrocellulose membranes and probed with antibodies raised against GFP (1:1,000), G6PDH (1:1,000), Pho8 (1:1,000), or CPY (1:2,500). For secondary labeling, blots membranes were stained with horseradish peroxidase-labeled affinity purified polyclonal antibodies to mouse or rabbit (1:10,000). Please refer to manufacturer's website for antibody validation. Chemiluminescence of stained membranes was digitally imaged using a GE Amersham Imager 600 (GE Health Care, Piscataway, USA). Blots shown are best representatives of 3 or 4 biological replicates, each repeated at least twice (technical replicates). Band density was measured using ImageJ software.

### Live cell imaging

Live yeast cells were treated with FM4-64 to exclusively stain vacuole membranes as previously described [28]. To stimulate down-regulation of pHluorin- or GFP- tagged surface proteins, cells in SC medium were treated with or without either 0.2 mM methionine for up to 30 minutes at 30°C or 100 μM cycloheximide for 90 minutes at 30°C, or incubated at 42°C for 30 minutes for sublethal heat stress [see 28]. After treatments, cells were immediately placed between pre-warmed glass coverslips at 30°C and imaging by HILO microscopy. For time-lapse videos, cells were plated on coverslips coated with concavalin-A (1 mg/ml in 50 mM HEPES, pH 7.5, 20 mM calcium acetate, 1 mM $MnSO_4$) and imaged at 30°C using a Chamlide TC-N incubator (Live Cell Instrument, Korea).

### Highly inclined laminated optical sheet (HILO) microscopy

Cross sectional images were acquired 1 μm into the sample using a Nikon Eclipse TiE inverted microscope equipped with a TIRF (Total Internal Reflection Fluorescence) illumination unit, Photometrics Evolve 512 EMCCD (Electron Multiplying Charge Coupled Device) or Photometrics Prime BSI sCMOS (scientific complementary metal–oxide–semiconductor) camera, Nikon CFI ApoTIRF 1.49 NA 100 X objective lens, and 488 nm or 561 nm 50 mW lasers operated with Nikon Elements software (Nikon Instruments Inc., Mississauga, Canada). Micrographs or movies shown are best representatives of 2–12 biological replicates (each represents a sample prepared on different days from a separate yeast culture), imaged at least 8 times each (technical replicates) whereby each field of view examined contained > 30 cells or isolated organelles.

### Yeast growth assay

Wild type or vps36Δ cells expressing Mup1-GFP were grown in 15 mL YPD medium overnight at 30°C. Cultures were diluted to 0.02 $OD_{600nm}$/mL in fresh YPD medium with or without 2.5 mM $MnCl_2$ and transferred to a clear flat-bottom 96-well plate (Corning Inc.). To measure growth, cultures were incubated at 30°C with continuous orbital shaking and

absorbance at 510 nm was measured at 5–minute intervals over 48 hours using a Tecan Sunrise microplate reader (Tecan, Switzerland). Data was background corrected using absorbance values of YPD medium alone and resulting growth curves show culture density recorded over time.

## Cell viability assay

As previously described [34], yeast cell cultures were grown in SC medium for 16–18 hours at 30˚C, sedimented, washed once with SC, resuspended in fresh SC, and incubated at either room temperature (control) or 50˚C (lethal heat stress) for 30 minutes. After incubation cells were washed once with fresh SC, resuspended in 100 μL SC and mixed with 100 μL 0.1% (w/v) methylene blue stock solution (0.1 g dissolved in 100 mL dH$_2$O), and incubated for 5 minutes at room temperature. Images were acquired using a Nikon Eclipse TiE inverted epifluorescence microscope equipped with a Nikon DsRi2 color CMOS (complementary metal-oxide semiconductor) camera, Nikon CFI 40 X Plan Apo Lambda 0.95 NA objective lens and DIC optics (Nikon Instruments Inc.). Viable cells were colorless and dead cells were blue. Cells were counted using ImageJ software (National Institutes of Health, Bethesda, USA) semi-automated cell counter macro. Data are reported as mean ± S.E.M of the percentage of blue cells observed within the population. Micrographs shown are best representatives of 2–4 biological replicates (each a single sample prepared from a separate yeast culture on different days), imaged at least 6 times each (technical replicates) whereby each field examined contained > 100 cells.

## Vacuole isolation and homotypic vacuole fusion

Vacuoles were purified from yeast cells as previously described [97]: Cells were harvested, washed, treated with oxalyticase to generate spheroplasts, and gently permeabilized using DEAE-dextran. Then vacuoles were isolated on a ficoll gradient by ultracentrifugation (100,000 x g, 90 minutes, 4˚C). Homotypic vacuole fusion reactions contained 6 μg of isolated vacuoles in standard fusion reaction buffer with 0.125 M KCl, 5 mM MgCl$_2$, 10 μM CoA and recombinant Pbi2 protein (a protease inhibitor). 1 mM ATP was added to initiate fusion. Vacuolar membranes were stained with 3 μM FM4-64 for 10 minutes at 27˚C. Reactions were incubated at 27˚C for up to 120 minutes and placed on ice prior to visualization by HILO microscopy. Where indicated, the fusion inhibitors 3.2 μM Gyp1-46 and 4 μM rGdi were added to block the reaction [68]. For heat stress treatment, isolated vacuoles were incubated at 37˚C for 5 minutes prior to adding them to reactions. Experimental data shown represent 3 or more biological replicates (each sample prepared from a separate yeast culture on different days) conducted in duplicate (technical replicates).

## pHluorin-based internalization assay

Ecliptic pHluorin (a pH-sentivie variant of GFP) was tagged to the cytoplasmic C-terminus of Mup1 protein [45]. As previously described [29], 30 μL fusion reactions containing 6 μg isolated vacuoles and standard reaction buffer were prepared on ice and then transferred to pre-warmed black 96-well conical-bottom microtiter plates. pHluorin fluorescence was recorded every 2 minutes for up to 90 minutes at 27˚C using a BioTek Synergy H1 multimode plate reading fluorometer (BioTek Instruments, Whiting, USA). Data shown are representative traces with values normalized to initial readings at 0 minutes and represent 3 biological replicates (each a sample prepared from a spate yeast culture on different days) conducted in duplicate (technical replicates).

### Data analysis and presentation

Movies and micrographs were processed using ImageJ and Adobe Photoshop CC software. Images presented were adjusted for brightness and contrast, inverted and sharpened with an unsharp masking filter. 3-dimensional or linear intensity profiles of GFP or FM4-64 fluorescence were generated using ImageJ software. Movie snapshots were selected to highlight docking and ILF formation during vacuole fusion. Group allocations were blinded for all micrographic analysis.

Cellular pHluorin or GFP location measurements were conducted using the ImageJ Cell Counter plugin. Micrographic data was quantified by counting total number of cells, as well as number of cells where the GFP fluorescence was detected on the plasma membrane, intracellular puncta, the vacuole membrane, or in the vacuole lumen. Values for each location were normalized to the total number of cells. Movies of vacuole fusion acquired in vivo were used to calculate the frequency of fusion events, reported as percentage of cells showing at least one event within 5 minutes; Only cells containing 2 or more vacuoles were counted.

Relative vacuole boundary, lumenal or outside membrane GFP fluorescence values were measured using ImageJ software as previously described [28]. Prior to quantification, micrographs were background subtracted and a 4x4 pixel region of interest was then used to measure mean GFP fluorescence on the boundary membrane, in the lumen or on the outer membrane of docked vacuoles only. Single vacuoles or docked vacuoles without a clear outer membrane (i.e. those in large clusters) were excluded. Mean GFP fluorescence intensity over area (boundary length, lumenal area or vacuole circumference) was calculated by measuring vacuole diameter (average of two lengthwise measurements) and boundary membrane length. Mean boundary GFP fluorescence was normalized to mean outer membrane fluorescence, to assess enrichment or depletion, and lumenal GFP fluorescence was normalized to total fluorescence (sum of boundary, outer membrane and lumen). Only cells or reactions containing clearly resolved docked vacuole membranes were used for micrographic analysis. To assess membrane fusion in vitro, we calculated vacuole surface area using averaged diameter measurements before ($t = 0$ minutes) and after addition of ATP to trigger fusion. Organelles were assumed to be spherical, and products of homotypic fusion are predicted to be larger than donor organelles.

When applicable, data are reported as column scatter plots as well as mean ± S.E.M. Comparisons were calculated using Student two-tailed $t$-tests; $P$ values are indicated and $P < 0.05$ suggests significant differences. Samples sizes were tested to ensure adequate power using online software (http://www.biomath.info). Variance was assumed to be similar between groups compared. Data was plotted using R-studio 3.6.3 or Synergy KaleidaGraph 4.0 software and Figs were prepared using Adobe Illustrator CC software.

## Supporting information

**S1 Fig. Confirmation of *vps36Δ* phenotypes.** (A) Wild type or *vps36Δ* cells expressing Mup1-GFP were grown in the absence (no stressor) or presence of 2.5 mM $MnCl_2$ and culture density was recorded over time. Traces shown are representations of 5 independent experiments. Growth of *vps36Δ* cells is sensitive to $MnCl_2$ as expected. (B) Western blot analysis of whole cell lysates prepared from wild type (WT) or *vps36Δ* cells expressing no GFP (BY 4741), Mup1-GFP or Mup1-pHluorin. Blots were stained for carboxypeptidase Y (CPY) or glucose-6-phosphate dehydrogenase (G6PDH; as load controls). Smaller mature (cleaved, m-CPY) and larger precursor forms of the enzyme (p1-CPY, p2-CPY), and estimated molecular weights are shown. (C) Mean steady–state m-CPY:p-CPY band density ratios were calculated, normalized to corresponding G6PDH band densities, and are shown for *vps36Δ* relative to isogenic wild

type (WT) for each strain genotype shown in B. n = 3 for each strain tested. *vps36Δ* strains tested show relatively low amounts of m-CPY as expected.
(TIF)

**S1 Video. Mup1 is present on ILFs formed during vacuole fusion in *vps36Δ* cells after methionine addition.** Time-lapse video showing a homotypic vacuole fusion event within a live *vps36Δ* cell expressing Mup1-pHluorin stained with FM4-64 10 minutes after methionine addition.
(MOV)

**S2 Video. Mup1 is present on ILFs formed during vacuole fusion in *vps27Δ* cells after cycloheximide addition.** Time-lapse video showing a homotypic vacuole fusion event within a live *vps27Δ* cell stained with FM4-64 expressing Mup1-GFP treated with cycloheximide.
(MP4)

**S3 Video. Mup1 is present on ILFs formed during vacuole fusion in *vps27Δ* cells after heat stress.** Time-lapse video showing a homotypic vacuole fusion event within a live *vps27Δ* cell stained with FM4-64 expressing Mup1-GFP after exposure to heat stress.
(MP4)

**S4 Video. Ste3 is present on ILFs formed during vacuole fusion in *vps27Δ* cells after heat stress.** Time-lapse video showing a homotypic vacuole fusion event within a live *vps27Δ* cell stained with FM4-64 expressing Ste-GFP after exposure to heat stress.
(MP4)

## Acknowledgments

We thank Beverley Wendland and Derek Prosser for yeast strains. Joshua Oliver imaged *vps36Δ* cells expressing Fet5-GFP shown in Fig 2C. C.K.G. is co-supervised by Alisa Piekny and is a fellow of the NSERC CREATE Training Program in Synthetic Biology Applications. We thank the Canadian Foundation for Innovation and Natural Sciences and Engineering Research Council of Canada for generous support of the Centre for Microscopy and Cellular Imaging at Concordia University.

## Author Contributions

**Conceptualization:** Charlotte Kathleen Golden, Erin Kate McNally, Joël Denis Richard, Christopher Leonard Brett.

**Formal analysis:** Charlotte Kathleen Golden, Thomas David Daniel Kazmirchuk, Erin Kate McNally.

**Funding acquisition:** Christopher Leonard Brett.

**Investigation:** Charlotte Kathleen Golden, Thomas David Daniel Kazmirchuk, Erin Kate McNally, Mariyam El eissawi, Zeynep Derin Gokbayrak, Joël Denis Richard.

**Methodology:** Charlotte Kathleen Golden, Erin Kate McNally, Christopher Leonard Brett.

**Project administration:** Christopher Leonard Brett.

**Resources:** Christopher Leonard Brett.

**Supervision:** Christopher Leonard Brett.

**Validation:** Charlotte Kathleen Golden, Thomas David Daniel Kazmirchuk, Erin Kate McNally, Zeynep Derin Gokbayrak.

**Visualization:** Charlotte Kathleen Golden, Thomas David Daniel Kazmirchuk, Erin Kate McNally, Zeynep Derin Gokbayrak, Christopher Leonard Brett.

**Writing – original draft:** Charlotte Kathleen Golden, Erin Kate McNally, Joël Denis Richard, Christopher Leonard Brett.

**Writing – review & editing:** Zeynep Derin Gokbayrak, Christopher Leonard Brett.

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
