## [Decision Letter · Decision Letter 0]

15 May 2022

Dear Dr Brett,

Thank you very much for submitting your Research Article entitled 'A Two–Tiered System for Selective Receptor and Transporter Protein Degradation' to PLOS Genetics.

The manuscript was fully evaluated at the editorial level and by independent peer reviewers. The reviewers appreciated the attention to an important problem, but raised some substantial concerns about the current manuscript. Based on the reviews, we will not be able to accept this version of the manuscript, but we would be willing to review a much-revised version. We cannot, of course, promise publication at that time.

Specifically, Reviewer #1 is enthusiastically positive about the manuscript, as is Reviewer #2, but the latter has some important critiques that will need to be addressed in a revision.

If you decide to revise the manuscript for further consideration at PLOS Genetics, please aim to resubmit within the next 60 days, unless it will take extra time to address the concerns of the reviewers, in which case we would appreciate an expected resubmission date by email to plosgenetics@plos.org.

[LINK]

We are sorry that we cannot be more positive about your manuscript at this stage. Please do not hesitate to contact us if you have any concerns or questions.

Yours sincerely,

Gregory P. Copenhaver

Editor-in-Chief

PLOS Genetics

Reviewer's Responses to Questions

**Comments to the Authors:**

Reviewer #1: In this study the Brett lab continues to reveal the intricacies of the ILF degradation pathway. Specifically, they show that the ILF pathway can serve as a second level of degradation for proteins that normally go through the ESCRT-dependent multvesicular body path. They followed the trafficking of the methionine transporter Mup1 in wild type cells and those with deletions in the in the ESCRT machinery. Mup1 is internalized upon the addition of Met and is trafficked to the vacuole as intraluminal vesicle cargo of multivesicular bodies. Upon fusion Mup1 is directly dilieverd to the lumen of the vacuole where it is degraded. When ESCRT was disabled Mup1 trafficked to the limiting membrane of the vacuole and was subsequently sorted to the boundary domain of docked vesicles. When vacuoles fuse together the Mup1 in the boundarly domain was interanalized as an ILF and degraded. This clearly shows that the ILF pathway is not only independent of the ESCRT machinery, but can also serve as a backup to ensure that membrane proteins are degraded when no longer needed.

This study was executed very well with the proper controls and their conclusions are supported by the data. I believe this paper will be of high impact in the field and suggest that it published as is.

Reviewer #2: Review is uploaded as an attachment.

**Have all data underlying the figures and results presented in the manuscript been provided?**

Reviewer #1: Yes

Reviewer #2: Yes

PLOS authors have the option to publish the peer review history of their article (what does this mean?). If published, this will include your full peer review and any attached files.

Reviewer #1: No

Reviewer #2: No

---

## [Decision Letter · Decision Letter 1]

26 Sep 2022

Dear Dr Brett,

We are pleased to inform you that your manuscript entitled "A two–tiered system for selective receptor and transporter protein degradation" has been editorially accepted for publication in PLOS Genetics. Congratulations!

Yours sincerely,

Gregory P. Copenhaver

Editor-in-Chief

PLOS Genetics

Comments from the reviewers (if applicable):

Reviewer's Responses to Questions

**Comments to the Authors:**

Reviewer #2: I would like to commend the authors for their excellent experimental responses to my initial review; I know that this was a lot of work over this time period. In fact, I almost feel a little guilty about asking for the vps36∆ vam3∆ strain- I'm sure that strain isn't very happy to work with!

Nevertheless, the addition of the vacuole fusion mutants to this revision - as well as the incorporation of the Yang et al. paper discussion - has made this work strong and convincing. While I might have anticipated that the vps36∆ vam3∆ strain might be far more sick after heat shock (especially more sick than the vam3∆ alone), these are the data. Importantly, the new in vivo work strongly supports an active role of vacuole fusion in ILF-mediate protein turnover (as expected).

Despite the continued apparent discrepancies regarding the role of the yeast ILF- and ESCRT-dependent protein degradation pathways recently presented from the Brett and Li laboratories, I look forward to (hopefully!) watching and reading about the future of this field. All of my concerns have been addressed in this resubmission, and I therefore enthusiastically recommend publication.

**Have all data underlying the figures and results presented in the manuscript been provided?**

Reviewer #2: Yes

PLOS authors have the option to publish the peer review history of their article (what does this mean?). If published, this will include your full peer review and any attached files.

Reviewer #2: **Yes: **Vincent J. Starai

**Data Deposition**

http://datadryad.org/submit?journalID=pgenetics&manu=PGENETICS-D-22-00373R1

**Press Queries**

---

## [Editor Report · Acceptance letter]

4 Oct 2022

PGENETICS-D-22-00373R1 

A two–tiered system for selective receptor and transporter protein degradation 

Dear Dr Brett, 

We are pleased to inform you that your manuscript entitled "A two–tiered system for selective receptor and transporter protein degradation" has been formally accepted for publication in PLOS Genetics! Your manuscript is now with our production department and you will be notified of the publication date in due course.

With kind regards,

Anita Estes

PLOS Genetics

On behalf of:
